# Effect of Dietary Phenolic Compounds on Incidence of Type 2 Diabetes in the “Seguimiento Universidad de Navarra” (SUN) Cohort

**DOI:** 10.3390/antiox12020507

**Published:** 2023-02-17

**Authors:** Zenaida Vázquez-Ruiz, Miguel Ángel Martínez-González, Facundo Vitelli-Storelli, Maira Bes-Rastrollo, Francisco Javier Basterra-Gortari, Estefanía Toledo

**Affiliations:** 1Department of Preventive Medicine and Public Health, Instituto de Investigación Sanitaria de Navarra (IdiSNA), University of Navarra, 31008 Pamplona, Spain; 2Biomedical Research Network Centre for Pathophysiology of Obesity and Nutrition (CIBEROBN), Carlos III Health Institute, 28029 Madrid, Spain; 3Department of Nutrition, Harvard T. H. Chan School of Public Health, Harvard University, Boston, MA 02115, USA; 4Grupo de Investigación en Interacciones Gen-Ambiente y Salud (GIIGAS), Instituto de Biomedicina (IBIOMED), University of León, 24004 León, Spain; 5Department of Endocrinology and Nutrition, Instituto de Investigación Sanitaria de Navarra (IdiSNA), Hospital Universitario de Navarra, 31008 Pamplona, Spain

**Keywords:** phenolic compounds, diet, diabetes, stilbenes, cohort

## Abstract

The global incidence of type 2 diabetes (T2D) has been steadily increasing in recent decades. The Mediterranean dietary pattern has shown a preventive effect on the risk of T2D. Evaluating the association between bioactive compounds such as phenolic compounds (PC) in a Mediterranean cohort could help to better understand the mechanisms implicated in this protection. We evaluated the association between dietary intake of PC and the risk of T2D in a relatively young cohort of 17,821 Spanish participants initially free of T2D, through the University of Navarra Follow-up Project (“Seguimiento Universidad de Navarra” or SUN cohort) after 10 years of median follow-up using time-dependent Cox models. Intake of PC was estimated at baseline and repeatedly at 10-year follow-up using a 136-item validated food frequency and the Phenol-Explorer database. The incidence of T2D was identified by a biennial follow-up, and only medically confirmed cases were included. During 224,751 person-years of follow-up, 186 cases of T2D were confirmed. A suboptimal intake of stilbenes was independently associated with a higher risk of T2D in subjects over 50 years (HR: 1.75, 95% CI: 1.06–2.90, *p* value < 0.05) after adjusting for potential confounders. Our results suggest that a moderate-high intake of stilbenes can decrease the risk of developing T2D in subjects over 50 years in our cohort.

## 1. Introduction

Diabetes is a chronic disease that is characterized by sustained high blood glucose levels which, over time, can lead to serious damage to the cardiovascular system, eyes, kidneys, and nerves. The number of people with diabetes nearly quadrupled from 1980–2014 with a faster increase in low and middle-income countries. Between 2000 and 2019 age-standardized diabetes mortality rates increased by 3%, and 1.5 million annual deaths are directly caused by diabetes [1], which implies a substantial increase in human and financial public cost [2]. The most common form of diabetes is type 2 diabetes (T2D), which usually appears in adulthood when the body becomes insulin resistant and the beta cells are unable to adapt to this reduced insulin sensitivity. Several studies have shown that the onset of T2D may be preventable in high-risk populations by modifying lifestyle factors such as adopting a healthy diet, maintaining a normal weight, exercising regularly and avoiding tobacco use [3,4,5].

A network analysis of nine dietary approach trials of >12 weeks duration showed decreases in HbA1c with all approaches [6], but greater glycemic benefits were seen with the Mediterranean dietary pattern and with a low-carbohydrate diet. However, the greater glycemic benefits of low-carbohydrate diets at 3 and 6 months ceased to be evident with a longer follow-up [7]. Trials of >6 months duration conducted in Mediterranean countries showed that adherence to the Mediterranean diet significantly contributes to a reduction in body weight and HbA1c levels, postponing the need for diabetes medication and providing cardiovascular health benefits [8]. The PREDIMED trial also provided strong evidence that a plant-based Mediterranean diet rich in phenolic compounds (PC) beneficially affects insulin sensitivity, inflammation, and oxidative stress in an older population (mean age: 67 years) [9]. Assessment of PC intake in a younger population with lower adherence to a Mediterranean diet, such as the SUN cohort, may be relevant and useful to better understand the role of PC intake on the risk of developing T2D in a Mediterranean country. Similar benefits on glycemic control have been ascribed to vegan and vegetarian diets [10]. Therefore, not only carbohydrate intake or glycemic index are determinants of the risk of developing T2D when dietary characteristics are considered, but also specific components present in plant-based dietary patterns may exert additional independent beneficial effects.

One of the most remarkable features of plant-based dietary patterns is the high concentration of PC. These bioactive compounds have an aromatic ring carrying at minimum one hydroxyl group, and their molecular structure can range from a single molecule to a high-molecular-weight complex polymer [11,12]. These compounds have been widely investigated and suggested to play an important role in preventing chronic diseases [13,14]. Oxidative stress is closely related to chronic disease, given that homeostatic redox imbalances can affect biomacromolecules and modify important proteins, promoting the pathogenesis of non-communicable diseases [15]. This accumulation of damaged molecules and impaired mitochondrial function results in a highly oxidative stress state, which is related to an increased incidence of age-related diseases such as cardiovascular disease (CVD), T2D, cancer and other chronic diseases [16,17,18]. As antioxidants, PC may protect cellular components against oxidative attack and, consequently, limit the risk of several degenerative diseases related to oxidative stress by their ability to trap free radicals and by modification of signal transduction that may enhance the up-regulation of antioxidant genes and the induction of endogenous antioxidant enzymes, such as superoxide dismutase, glutathione peroxidase, glutathione or catalase [19].

A recent review of epidemiological evidence from different countries of Europe, Asia and America, showed that six prospective cohorts and nine randomized controlled trials observed antidiabetic effects of PC [20]. In those studies, food frequency questionnaires (FFQ) and Phenol-Explorer database were used to determine dietary intake of PC. Nevertheless, sources of PC are likely to be different in Mediterranean and in non-Mediterranean countries [21]. In the traditional Mediterranean dietary pattern, extra virgin olive oil, nuts, red wine and olives are excellent sources of PC, and these foods may have an important role to play in T2D prevention [18,22]. Moreover, a wide variability in intake may be found when the consumed amount of these foods and beverages is high. Therefore, our objective was to longitudinally evaluate the association between total PC intake and by specific classes of PC on the risk of developing T2D in a large Mediterranean cohort, the “Seguimiento Universidad de Navarra” (SUN). Benefits attributed to long-term PC intake are best assessed in cohorts with prolonged follow-up, a high retention rate and well-controlled confounders.

## 2. Materials and Methods

### 2.1. Study Population

The SUN project is a prospective, dynamic, and multipurpose cohort of Spanish university graduates, recruitment for which began in December 1999. As of May 2022, 23,133 Spanish university graduates were participating. A detailed description of design and methodology of this cohort is available elsewhere [23]. In summary, self-reported questionnaires returned by mail or online at the beginning of the study collected information on health status, family history of disease, physical activity, socio-demographic characteristics, and other aspects of the participants’ lifestyles. Updated information on lifestyle factors, especially dietary habits and health outcomes, is collected every two years through repeated follow-up questionnaires. The Institutional Review Board of the University of Navarra endorsed the study protocol and admitted that voluntary responses to the first questionnaire were sufficient to be considered as a valid informed consent to participate in the study.

For these analyses, participants with less than 2 years and 9 months of follow-up (*n* = 236) were excluded, together with those with total energy intake outside the predefined limits [24] (<500 or >3500 kcal/d in women, or <800 or >4000 kcal/d in men) (*n* = 2123). We also excluded participants with >15 items missing in the food frequency questionnaire (FFQ) (*n* = 1212). In addition, participants with any type of diabetes or those who underwent pancreatectomy were excluded (*n* = 377), as were those with questionable information on the additional diabetes confirmation questionnaire (*n* = 2). Finally, a total of 17,821 participants were examined, with a mean follow-up time of 12.6 years (SD = 5.7) and a retention rate of 92.8% (Figure 1).

### 2.2. Assessment of Food and PC Intake

A semi-quantitative FFQ, previously validated with 136 food records, was used to evaluate dietary intake at both study inclusion and at 10-year of follow-up [25,26,27]. For each food item, 9 options of intake were provided in the FFQ (>6 times/day; 4–6 times/day; 2–3 times/day; once/day; 5–6 times/week; 2–4 times/week; once/week; 1–3 times/month and never or almost never). To calculate the daily intake of nutrients and energy, the standard servings were multiplied by the frequency of consumption of each food and by the nutritional content or kilocalories of energy specified in Spanish food composition tables [28,29]. Other eating habits such as snacking between meals or following a special diet were also gathered at baseline. Compliance with the Mediterranean diet was measured with the score (0–9) of Trichopoulou et al. [30].

The Phenol-Explorer database version 3.6 (www.phenol-explorer.eu (accessed on 15 December 2022), whose methodology has been published elsewhere, was used to calculate the PC content of each food [31,32]. The most common extraction method used for estimating PC intake was high-performance liquid chromatography (HPLC), which simultaneously quantifies phenolic acid esters and phenol glycosides together with aglycones and free phenolic acids, and normal-phase HPLC to estimate the proanthocyanidins content. Data from HPLC after hydrolysis method were used, if available, to calculate the aglycone in phenolic acids and lignans in some foods such as olives, beans and grains, since this analytical method uses hydrolysis to release certain compounds which cannot be solubilized without hydrolysis. Foods with only traces of PC were excluded. A weighted average was calculated according to the average dietary consumption of the Spanish adult population when the FFQ items contained more than one food [33], and for processed foods and recipes PC content was calculated according to their ingredients. Finally, the PC intake from each food was calculated by multiplying the content of each PC by the daily consumption of each food. The total PC intake (milligrams per day) was determined as the sum of all PC of each food item reported in the FFQ, and was also calculated by class intake. Additionally, we calculated the contribution of each specific food or food group to the total PC intake and expressed it as percentage.

### 2.3. Outcome Assessment

Participants with a self-reported medical diagnosis of diabetes or regular use of oral antidiabetics or insulin at baseline were classified as prevalent cases of diabetes. A self-reported clinical diagnosis of T2D during any follow-up questionnaire was considered a probable incident case; these participants were sent an additional specific questionnaire to confirm their diagnosis and to collect more detailed information such as date of diagnosis, type of diabetes, glycosylated hemoglobin (HbA1c) above 6.5%, number of times with a fasting glucose value above 126 mg/dL, glucose 2 h after an oral glucose tolerance test above 200 mg/dL and use of hypoglycemic drugs. These probable cases were asked for medical reports detailing their diagnosis and an endocrinologist, blinded to the dietary and lifestyle variables, evaluated these T2D medical reports and adjudicated confirmed cases according to the criteria of the American Diabetes Association [34]. Detailed information on the methods of T2D case ascertainment in the SUN cohort has been published elsewhere [35].

### 2.4. Assessment of Covariates

Participants reported information at baseline on potential confounders, including validated anthropometric measures (height and weight) [36], smoking status, TV watching, leisure-time physical activity (METs-h/week) [37], socio-demographic characteristics (age, sex, marital status and level of education) and other health-related behaviors. Personal and family medical history (hypercholesterolemia, hypertension, CVD, hormone replacement therapy or cancer among others) was also provided at the beginning of the study. The reliability of self-reported medical data was assessed by some validation studies [38,39].

### 2.5. Statistical Analysis

Total and class-specific intake of PC were adjusted for total energy intake according to the residuals method suggested by Willett [24] and categorized into sex-specific quintiles. Descriptive statistics were performed to examine participants’ baseline information adjusted for age and sex according to the inverse probability weighting method, using proportions or means (and standard deviations, SDs) based on quintiles of PC intake. The main sources of total PC intake and classes were calculated, as well as the contribution of individual foods to total PC intake and classes. 

Hazard ratios (HR) were estimated using crude and time-dependent multivariable Cox regression models to analyze the risk of developing T2D and their 95% confidence intervals (CIs), considering age as the underlying time variable and the first quintile as the reference category. Time at entry was determined as the date of completion of the baseline questionnaire and the time of exit as the date of the last follow-up questionnaire, or diagnosis of T2D, whichever occurred first. Multivariable adjusted model 1 was stratified by age (five-year periods), marital status, years of college education and recruitment period, and adjusted for total energy intake (kcal/day), body mass index (kg/m^2^), and its quadratic term to account for departures of linearity, smoking status (never smoker, current smoker or former smoker), lifetime tobacco exposure (packs per year), hypertension, dyslipidemia, CVD, family history of diabetes, physical activity (METs-h/week), health consciousness (number of medical check-ups), TV watching (hours/day), energy-adjusted alcohol intake from sources other than wine (g/day) and energy-adjusted glycemic index (0–100). Repeated measurements analyses were performed on participants who filled out the FFQ after 10-year follow-up (Q10 questionnaire), using cumulative averages to weaken the effect of dietary modifications between baseline and 10 years of follow-up and stratifying and adjusting for the same confounders. HRs in model 1 and repeated measurements were also determined, restricting the analysis to subjects older than 50 years. This was an a priori decision based on previously published data on the incidence of T2D in Spain [40]. We restricted the analysis to older participants to accommodate the assessment to the higher risk of diabetes that they present. In addition, dietary exposures probably need a long induction period to contribute to causing diabetes, and assessing this association at younger ages does not allow us to capture this longer induction period. Tests of linear trend were performed, assigning to each quintile its median and using the resulting variable as continuous in the aforementioned models. In addition, HRs were calculated to assess the effect of a suboptimal PC intake (first quintile of total PC intake) versus a reference category of moderate–high intake (quintiles 2 to 5 of PC intake merged in a single group). All analyses were repeated for each PC class. Between-person variability of specific classes of PC intake was calculated for those in which the results were statistically significant. Interactions between PC intake and BMI or sex were also explored (a priori decisions). No other interactions were evaluated.

Sensitivity analyses were also conducted to examine the robustness of our results, excluding T2D diagnosed in the first 2 years of follow-up. Data analyses were performed with the Stata v.17 statistical software package (College Station, TX, USA; StataCorp LLC). The *p*-values presented are all two-tailed and statistical significance was set at 0.05.

## 3. Results

Table 1 presents the main baseline characteristics of the 17,821 participants included in the analyses by quintiles of baseline energy-adjusted PC intake. Participants’ mean age was 37.4 years (SD = 11.8) and the mean BMI was 23.4 kg/m^2^ (SD = 3.5).

Participants in the last PC intake quintile showed the highest percentage of women and married people and a higher mean age. Additionally, in this quintile participants showed higher physical activity (MET-h/week) and a higher percentage of energy provided by carbohydrates, as well as higher consumption of fiber, alcohol and better adherence to the Mediterranean diet than the other quintiles. However, participants in the first quintile showed higher total caloric intake and a slightly higher percentage of smokers.

The energy-adjusted mean ± SD of total PC intake was 784.1 ± 335 mg/day, of which 55.5% were flavonoids (435.7 ± 237.7 mg/day), 38.8% phenolic acids (304.1 ± 152.4 mg/day), 5% other PC (40.9 ± 33.7 mg/day), 0.28% lignans (2.2 ± 0.96 mg/day), and 0.15% stilbenes (1.24 ± 2.63 mg/day) (Table 2). By class, flavonoids contributed the most to total PC intake, followed by phenolic acids, but according to subclasses, hydroxycinnamic acids (34.1%) and proanthocyanidins (28.4%) contributed the most to the PC intake.

Regarding PC subclasses, proanthocyanidins were the main contributors to the flavonoids (222.6 ± 199 mg/day), and chocolate was the principal food source (43.8%) followed by apple (22%) and cherries (11.1%). The second contributor to flavonoids was flavanones (75.1 ± 76.5 mg/day), which were mostly provided by oranges (43.4%) and natural orange juice (34.4%). Among phenolic acids, hydroxycinnamic acids showed the highest intake (267 ± 140.3 mg/day) with coffee (33.3%) as the main source. Olives were practically the only food source of hydroxyphenylacetic acids (4.5 ± 7.1 mg/day) and also for hydroxyphenylpropanoic acids (0.6 ± 0.9 mg/day). Lignans intake (2.2 ± 0.96 mg/day) was mostly provided by carrot and pumpkin (12.3%) and olive oil (11.4%). The primary source of stilbenes (1.24 ± 2.6 mg/day) was red wine (89.6%). Tyrosol intake (27.4 ± 30.5 mg/day) mostly provided by olives (64.9%) and olive oil (29.7%) was the most abundant PC within the subclass of “other phenolic compounds”.

A total of 186 new medically confirmed cases of T2D were identified among 224,747 person-years of follow-up (median follow-up of 12.6 years). Analysis comparing a low or suboptimal (Q1) vs. a moderate-high (Q2–Q5) intake were also fitted for total and by class PC intake (Table 3). Participants with a low total PC intake showed a non-significant higher risk of developing T2D in the multivariable model 1, both with the baseline intake information (1.21; 95% CI 0.74–1.95; *p*-value: 0.4) and with repeated measurements (HR: 1.29; 95% CI 0.80–2.09; *p*-value: 0.3). Nevertheless, when analyses were limited to participants over 50 years old, lower HRs were observed. Analyzing the flavonoid class separately, in the repeated measurement analysis restricted to those older than 50 years an increased risk was observed for those with low/suboptimal (Q1 vs. Q4–Q5) intake (HR: 1.52; 95% CI: 0.95–2.44); *p*-value: 0.08), although it did not reach statistical significance in any regression model. For a low/suboptimal lignan intake, higher but non-significant risks were observed in a low/suboptimal intake (Q1 vs. Q4–Q5) when analyses were limited to participants over 50 years in the multivariate model 1 (HR: 1.53; 95% CI: 0.95–2.45; *p*-value: 0.07) and repeated measurements (HR: 1.52; 95%CI: 0.96–2.40; *p*-value: 0.07). As for phenolic acids, a slight non-significant decrease in the risk of developing diabetes was observed in model 1 (HR: 0.75; 95% CI: 0.43–1.32; *p*-value:0.33) and in repeated measurements (HR: 0.75; 95% CI: 0.43–1.33; *p*-value:0.32) for the comparison of Q1 vs. Q4–Q5 of phenolic acids intake, but this effect was almost lost in both models once we restricted the analysis to subjects over 50 years of age. A low stilbenes intake showed higher risks of developing T2D in repeated measurements model restricted to those participants aged over 50 years (HR: 1.75; 95% CI: 1.06–2.90; *p*-value: 0.03), and were the only PC class in which a significantly increased risk of developing T2D was observed. Regarding a low intake of other PC classes, the risk of a low/suboptimal intake showed unclear and nonsignificant results that may be due to the heterogeneity of this group of PC.

Cumulative R^2^ values were calculated from nested regression analyses after stepwise selection to calculate between-person variability of stilbenes intake, and red wine contributed most to its total between-person variability in our study. No significant linear trend was found between total or class-specific PC intake quintiles and the possible interactions between PC intake and BMI or sex were not statistically significant.

The abovementioned sensitivity analyses yielded similar results.

## 4. Discussion

These results showed that low/suboptimal intake (0.6 mg/day or less) of stilbenes was associated with an increased risk of T2D at 10-year follow-up, and significant results were observed in participants older than 50 years with a 75% relative increased hazard of T2D. In addition, although the results were not significant, an increased risk of T2D at 10-year follow-up was observed for a low/suboptimal total PC intake (415.9 mg/day or less) as well as for low intake of flavonoids (203.9 mg/day or less) and lignans (1.6 mg/day or less), both in the total study sample and in those participants aged over 50 years. However, a decrease in the risk of developing T2D at low phenolic acid consumption (187.7 mg/day or less) was observed, although the results were not significant. In the subgroup of other PC intake, no relevant association was observed, probably due to the heterogeneity of its constituent.

In our cohort the main food source of stilbenes was red wine, and it was also the main source of between-person variability. Resveratrol was the main PC contributing to stilbenes followed by piceatannol, a metabolite of resveratrol. Therefore, the risk reduction in developing T2D for stilbenes consumption could not be dissociated from alcohol consumption in our cohort. Wines were nearly equal in alcohol and caloric content, but the levels of total PC in red wine were 7-fold higher than white wine and 2.5-fold higher in rose wine than in white wine [31]. The FFQ in our study collected separately red wine and other types of wine (rosé and white). A 2-year randomized controlled trial that assessed the effects of initiating a moderate alcohol intake in adults with well-controlled T2D on cardiometabolic risk—the CASCADE trial—observed that the effect of wine on glycemic control was mainly driven by alcohol, and pointed out the role of genetic interactions in glucose metabolism since slow ethanol metabolizers significantly benefited from the effect of red wine on some biomarkers of glycemic control as fasting plasma glucose, HOMA-IR, and hemoglobin A1c compared with fast ethanol metabolizers. However, a stronger effect of red wine was seen on lipids levels and overall variables of the metabolic syndrome [41]. Additionally, in a cross-over trial including 67 men at high cardiovascular risk designed to compare the effects of a moderate intake of red wine (30 g alcohol/day), the equivalent amount of dealcoholized red wine and gin (30 g alcohol/day) on glucose metabolism and lipid profile (during 4 weeks) showed that red wine with or without alcohol, but not gin, improved glucose metabolism as measured by HOMA-IR [42]. Other clinical trials have demonstrated improvements in insulin sensitivity with the consumption of red wine for 2 weeks without affecting vascular reactivity and nitric oxide production [43] and also with red grape juice consumption [44], but the results are inconsistent [45,46]. Positive effects on glucose metabolism have been observed in intervention studies with grape PC supplementation [47,48].

In our study, an increased risk of T2D was observed for low intakes (Q1 vs. Q2–Q5) of flavonoids and lignans at 10 years, especially in participants older than 50 years, but these results were not significant. A study of three large US cohorts showed that higher consumption of anthocyanins, a flavonoid subclass, was associated with a lower risk of T2D in adults who were free of diabetes, CVD, and cancer at baseline [49], while in the same line a large European cohort, the EPIC-cohort, showed an inverse association between total flavonoid intake, particularly flavanols and flavonols subclasses, and T2D but not with lignans [50]. However, lignans have demonstrated in other studies the potential of lignan-rich diets against the development of various diseases and diabetes [51,52]. Dietary fiber may be responsible, in part, for the associations observed with lignan intake and insulin resistance [53]. Lignan intake was mainly contributed by carrot, pumpkin, and olive oil in our study.

Regarding phenolic acids, most came from coffee in our study, where results showed a non-significant reduction on the risk of developing T2D with a suboptimal intake. Evidence on insulin resistance and phenolic acids is controversial. An 8-week trial to address the effect of the consumption of five cups/day of coffee among healthy subjects found no changes in insulin or glucose markers [54], but other studies showed benefits [55]. Other phenolic compound classes showed unclear results in all models analyzed in our study. Half of other PC classes were tyrosols, which mainly come from olives and olive oil and have been shown to have a beneficial effect in T2D prevention [56]. Protective effects of total PC intake were not as evident as those of the flavonoids, lignans and stilbenes at 10 years in the over-50s, probably because the results for the other PC class were unclear and phenolic acids showed an opposite, but also non-significant, effect that may have affected results for total PC intake. Some studies suggest that PC-rich diets show beneficial effects on glucose homeostasis [57,58], even using total PC urinary excretion [59]. According to our results and the available evidence, it seems that the protective effects are stronger in diets rich in some classes of PC such as flavonoids, lignans or stilbenes.

Epidemiological, randomized, controlled dietary studies in humans or animal models provide ample evidence that dietary PC intake improves diabetes risk by regulating key pathway of carbohydrate metabolism and hepatic glucose homeostasis, including glycolysis, glycogenesis, and gluconeogenesis. The inhibition of intestinal glycosidases and glucose transporter by PC has been studied, and some PC decrease S-Glut-1 mediated intestinal transport of glucose [60]. Moreover, PC themselves may directly inhibit the development of inflammation and suppress appetite by stimulating the secretion of CCK and leptin, so they may have some anti-obesogenic effect and, consequently, play an anti-diabetic effect to some extent [20]. Moreover, the antioxidant and anti-glycation functions of PC can reduce inflammation caused by reactive oxygen species (ROS) and advanced glycation end products (AGEs), thus protecting pancreatic β-cells [61]. The stilbene resveratrol has been reported to act as an anti-diabetic agent. Many mechanisms have been proposed to explain the anti-diabetic action of this stilbene. As such, stilbene-mediated modulation of SIRT1 has been observed to improve whole-body glucose homeostasis and insulin sensitivity in diabetic rats [62,63]. In addition, stilbene has also been shown to induce antioxidant enzymes such as catalase, superoxide dismutase and hemoxygenase-1 and suppress the expression of inflammatory biomarkers such as TNF, COX-2, iNOS and CRP [64].

### Strengths and Limitations

This study presented some limitations. One of them may be linked to the computation of PC intake, and although the most recent and complete database available was used (Phenol-explorer version 3.6) not all FFQ-derived foods are included in the database (e.g., honey) and FFQ does not include all PC-rich foods (e.g., spices, herbs, or some seeds). Another aspect to consider is that FFQ does not discern foods in detail, which may affect the PC content of different foods. (cocoa percentage of different varieties of chocolate, types of coffee or tea). The potential measurement error more likely contributed to attenuate the estimates of relative risk that we presented here. Although at the moment we cannot biologically validate PC intake in our cohort, some validation studies have concluded that PC intake assessed by FFQ shows an adequate correlation with total urinary excretion of PC [65,66,67,68]. In addition, bioavailability after consumption was not taken into consideration and it may show inter-person variability. Moreover, PC food content can be influenced by external conditions such as storage, ripeness, processing and variety, which may affect the PC intake [69]. Furthermore, we have previously reported that the consumption of polyphenol-rich foods (according to the same FFQ that we used) was appropriately correlated with total polyphenol excretion in urine and with objectively measured biomarkers of cardiometabolic risk [66,67,68].

Fortunately for our participants, the number of cases in the SUN project was comparatively low as expected in a relatively young and healthy Mediterranean cohort. However, this fact may have limited our statistical power for evaluating these associations and the lack of significance in many of our estimates, also given that the inherent measurement errors may have contributed to an underestimation of the effects. This study also has several strengths, including comprehensive and validated data on possible confounders, prospective design, high retention rate of the cohort (92%) and ascertainment of events by a blinded endocrinologist. In addition, errors in dietary measurements were reduced by considering dietary changes at 10 years of follow-up, which allowed a better representativeness of dietary intake over time. This research was carried out in a Mediterranean country, which may be useful to properly capture the effect of PC-rich foods, such as olive oil or red wine that are traditionally included in the Mediterranean dietary pattern.

## 5. Conclusions

These findings support avoiding a suboptimal stilbenes intake in a population over 50 years old and promoting diets rich in stilbenes to reduce the risk of T2D. Stilbenes were found in greater abundance in red wine and grapes. Future research, especially randomized controlled trials, should monitor the confounding effect of culinary preparations. Furthermore, it could provide strong evidence to clarify the biological role of red wine components in the prevention of chronic diseases.

## Figures and Tables

**Figure 1 antioxidants-12-00507-f001:**
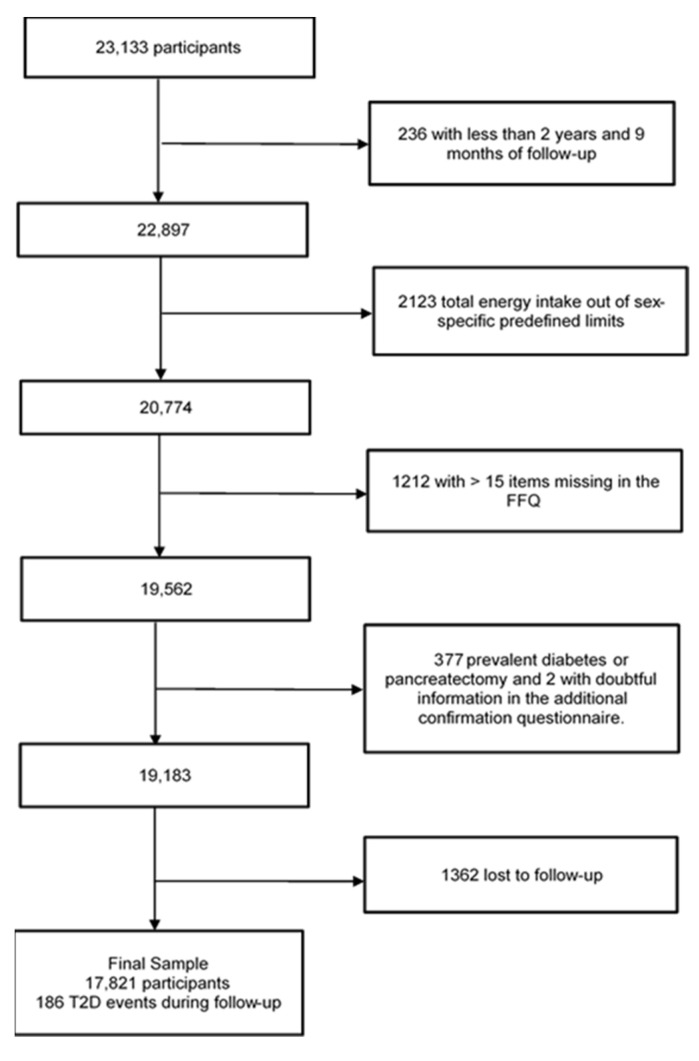
Flowchart of participants in the SUN (‘Seguimiento Universidad de Navarra’) included in analyses of dietary intake of phenolic compounds and incident type 2 diabetes. Abbreviation: Type 2 diabetes (T2D); Food frequency questionnaire (FFQ).

**Table 1 antioxidants-12-00507-t001:** Baseline characteristics * of participants across sex-specific energy-adjusted quintiles of total phenolic compounds dietary intake.

	Energy-Adjusted Quintiles of Total Phenolic Compounds Intake				
	Q1 (*n* = 3565)	Q2 (*n* = 3564)	Q3 (*n* = 3564)	Q4 (*n* = 3564)	Q5 (*n* = 3564)
Age (years)	32.8 (10.2)	35.89 (11.2)	37.67 (11.52)	39.46 (11.83)	41.4 (12.4)
Sex (% women)	56.1%	60.2%	62.2%	62.7%	63.9%
Total phenolic compounds (mg/day)	415.9 (98.3)	602.6 (39.6)	734.6 (39.0)	892.4 (55.8)	1279.4 (329.1)
Flavonoids (mg/day)	203.9 (86.1)	314.9 (78.4)	397.6 (91.4)	497.8 (117.1)	776.4 (312.8)
Lignans (mg/day)	1.6 (0.6)	1.9 (0.6)	2.2 (0.7)	2.4 (0.8)	2.9 (1.3)
Phenolic acids (mg/day)	182.7 (70.3)	250.2 (74.4)	293.9 (85.7)	345.7 (107.6)	442.7 (179.6)
Stilbenes (mg/day)	0.6 (1.2)	0.9 (1.7)	1.2 (2.1)	1.5 (2.7)	1.8 (3.8)
Other phenolic compounds (mg/day)	27.1 (19.1)	34.6 (19.9)	39.8 (21.8)	45.0 (27.6)	55.5 (51.7)
Body mass Index (kg/m^2^)	23.5 (3.6)	23.5 (3.4)	23.5 (3.5)	23.5 (3.4)	23.3 (3.5)
Physical activity (METS-h/week)	18.7 (21.5)	19.6 (20.5)	22.2 (22.5)	23.3 (23.1)	26.4 (27.4)
University education (years)	5.0 (1.5)	5.1 (1.5)	5.1 (1.6)	5.1 (1.5)	5.0 (1.5)
TV watching time (hours)	1.7 (1.3)	1.6 (1.2)	1.6 (1.1)	1.6 (1.1)	1.5 (1.1)
Smoking					
Current smoker	23.8%	22.9%	22.7%	23.7%	22.2%
Former smoker	25.7%	26.3%	28.6%	28.3%	29.7%
Never smoker	49.9%	50.1%	48.2%	47.3%	47.5%
Package-years of smoking	6.5 (11.5)	5.7 (9.6)	5.8 (9.5)	5.9 (9.4)	6.2 (9.7)
Marital status (% married)	37.8%	46.2%	52.2%	54.7%	58.1%
Dyslipidemia at baseline	15.4%	15.0%	17.2%	17.4%	18.1%
Hypertension at baseline	10.5%	10.3%	10.5%	9.4%	10.1%
Family history of diabetes	15.6%	15.9%	15.1%	15.0%	15.7%
CVD at baseline	1.4%	1.4%	1.4%	1.2%	1.5%
Hormone replacement therapy (% of women)	0.85%	1.5%	2%	2.15%	3.1%
Total energy intake (Kcal/day)	2580 (567.2)	2293 (584)	2236 (596)	2297 (592)	2470 (614)
Carbohydrate intake (% energy)	42.6 (7.5)	42.4 (6.9)	43.0 (6.7)	44.0 (6.8)	46.2 (7.5)
Protein intake (% energy)	17.9 (3.2)	18.5 (3.1)	18.6 (3.0)	18.4 (3.2)	17.7 (3.1)
Fat intake (% energy)	37.9 (6.6)	37.3 (6.0)	36.4 (5.9)	35.3 (6.0)	33.9 (6.6)
Dietary fiber intake (g/day)	19.7 (7.4)	19.8 (7.5)	21.3 (7.8)	23.7 (8.7)	29.5 (12.1)
Alcohol intake (g/day)	5.5 (9.3)	6.0 (8.6)	6.4 (8.8)	7.1 (9.9)	7.6 (11.6)
Adherence to MDS (0–9 score)	3.4 (1.6)	3.8 (1.6)	4.2 (1.7)	4.6 (1.7)	5.1 (1.7)
Glycemic Index (0–100)	54.2 (4.6)	53.0 (4.2)	52.2 (4.0)	51.5 (4.1)	49.9 (4.0)

* Adjusted for inverse probability weighting for sex and age, except for age and sex. Values are expressed as means and standard deviations or percentage. SUN: Seguimiento Universidad de Navarra; CVD: Cardiovascular disease; MET: metabolic equivalents; MDS: Mediterranean Diet Score proposed by Trichopoulou et al. [31].

**Table 2 antioxidants-12-00507-t002:** Contribution of phenolic compounds subclasses to total phenolic compounds intake and food sources.

Phenolic Compounds Classes and Subclasses	Mean (mg/d) ± SD, (%)	Food Sources * (% of Contribution)
Flavonoids	435.7 ± 237.7, (55.5)	
Anthocyanins	38.4 ± 44, (4.9)	Cherries (44.9), strawberries (15.7), red wine (14.8), grapes (13.3), olives (8.1), beans (1.3).
Chalcones	0.003 ± 0.006, (<0.01)	Beer (100).
Dihydrochalcones	1.95 ± 2.6, (0.25)	Apples (100).
Dihydroflavonols	1.42 ± 3.3, (0.18)	Red wine (98.3), other wines (1.7).
Flavan-3-ols	21.23 ± 16.1, (2.7)	Apples (27.6), chocolate (21.1), red wine (15.4), peaches (10.1), cherries (6.4), grapes (5.4), strawberries (2.9), green beans (2.5), banana (2.5) lentils (1.3).
Flavanones	75.1 ± 76.5, (9.5)	Oranges (43.4), natural orange juice (34.4), fruit juices from concentrate (18.8), tomato (1.4), other fruit juices (1.3).
Flavones	17.6 ± 13.3, (17.6)	Other vegetables (37.4), natural orange juice (17.6), cookies (10.1), olives (6), fruit juices from concentrate (4.6), chocolate cookies (3), watermelon (2.8), industrial bakery (2.3) pastries (2.2), croquettes (1.8), peppers (1.7), pizza (1.4), cupcake (1.1).
Flavonols	57.4 ± 36.2, (7.3)	Lettuce (36.3), Swiss chard leaves (29.1), asparagus (11.4), olives (3.2), nuts (3), green beans (2.7), cabbage (1.7), chocolate (1.6), tomato (1.5), apples (1.5).
Isoflavonoids	0.04 ± 0.3, (0.04)	Beans (69.6), nuts (27.7), beer (2.6).
Proanthocyanidins	222.6 ± 199.1 (28.4)	Chocolate (43.8), apple (22), cherries (11.8), strawberries (5.2), grapes (4.4), nuts (4.2), red wine (3.3), beans (3.2).
Lignans	2.2 ± 0.96, (0.3)	Carrot and pumpkin (12.3), olive oil (11.4), tomato (9), broccoli and cabbage (7.6), oranges (7), green beans (6), peaches (4.4), pepper (3.5), strawberries (3), asparagus (2.8), red wine (2.8), cold tomato soup (2.8), cookies (2.8), apples (2.3), grapes (1.8), eggplant zucchini and cucumber (1.7), kiwi (1.4), dried fruit (1.4), fried potatoes (1.3), melon (1.3), nuts (1.1).
Phenolic acids	304.1 ± 152.4, (38.8)	
Hydroxybenzoic acids	32 ± 25.3, (4.1)	Nuts (35), olives (16.5), strawberries (12.7), carrots and pumpkin (12.3), Swiss chard leaves (7.7), red wine (5.6), beer (2.6), apples (1.4).
Hydroxycinnamic acids	267 ± 140.3, (34.1)	Coffee (33.3), decaffeinated coffee (12.7), other vegetable (8.5) carrots and pumpkin (7.7) cherries (6.5), French fries (5.5), olives (4.7), apple (4.3), baked or boiled potatoes (3) tomato (1.85), peaches (1.4), nuts (1.4), orange juice (1.3) red wine (1.2).
Hydroxyphenylpropanoic acids	0.6 ± 0.9, (0.07)	Olives (100).
Hydroxyphenylacetic acids	4.5 ± 7.1, (4.5)	Olives (96.4), red wine (2.7).
Stilbenes	1.24 ± 2.6, (0.15)	Red wine (89.3), grapes (3.6), other wines (3.5), strawberries (2.3).
Other phenolic compounds	40.9 ± 33.7, (5.2)	
Alkylmethoxyphenols	0.3 ± 0.5, (0.03)	Decaffeinated coffee (84.3) beer (15.7).
Alkylphenols	8 ± 11.2, (1)	Breakfast cereals (43.4), whole-grain bread (41.5), pasta (8.2), cookies (2.3).
Furanocoumarins	0.4 ± 0.7, (0.05)	Other vegetables (100).
Hydroxybenzaldehydes	0.2 ± 0.43, (0.2)	Red wine (91), beer (2.7), other wines (2.6), olives (1.8), whisky (1.4).
Hydroxybenzoketones	0.001 ± 0.002, (<0.01)	Beer (100)
Hydroxycoumarins	0.05 ± 0.09, (<0.01)	Beer (75.4), other wines (24.6)
Methoxyphenols	0.03 ± 0.09, (<0.01)	Decaffeinated coffee (100)
Tyrosols	27.4 ± 30.5, (3.5)	Olives (64.9), olive oil (29.7), red wine (3.4), cold tomato soup (1.2).
Other phenolic compounds (subclass)	4.6 ± 6.5 (0.6)	Orange juice (66.7), other fruits juice (22.2), Coffee (6.1), apples (2.5), olives (1.5)

* Food sources that contribute more than 1%.

**Table 3 antioxidants-12-00507-t003:** Hazard ratio (HR) and 95% confidence intervals (CI) of confirmed type 2 diabetes cases according to quintiles of total and class-specific phenolic compounds intake.

	Energy-Adjusted Quintiles of Total Phenolic Compounds Intake						Low Intake vs. Medium–High Intake
	1 *n* = 3565	2 *n* = 3564	3 *n* = 3564	4 *n* = 3564	5 *n* = 3564	*p* for Trend	Q1 vs. Q2–Q5
**Total phenolic compounds**							
Median intake (mg/d)	434.9	602.6	733.4	887.8	1180.1		
Cases	29	32	36	41	48		
Person-years	45,183	45,561	45,243	44,585	44,180		
Age-sex adjusted HR (95% CI)	1 (Ref.)	0.82 (0.50–1.37)	0.80 (0.49–1.30)	0.77 (0.48–1.25)	0.77 (0.48–1.25)	0.39	1.26 (0.84–1.88)
Multivariable adjusted model 1	1 (Ref.)	1.03 (0.58–1.85)	0.81 (0.45–1.44)	0.85 (0.48–1.49)	0.68 (0.39–1.20)	0.12	1.21 (0.74–1.95)
Restricted to >50 years old	1 (Ref.)	1.26 (0.68–2.35)	0.87 (0.46–1.64)	1.05 (0.57–1.94)	0.85 (0.46–1.58)	0.44	1.00 (0.60–1.69)
Repeated measurements model 1	1 (Ref.)	0.93 (0.51–1.64)	0.72 (0.40–1.30)	0.83 (0.48–1.45)	0.67 (0.38–1.15)	0.15	1.29 (0.80–2.09)
Restricted to >50 years old	1 (Ref.)	0.88 (0.48–1.63)	0.95 (0.53–1.70)	0.92 (0.51–1.65)	0.80 (0.45–1.45)	0.56	1.12 (0.69–1.82)
**Flavonoids**							
Median intake (mg/d)	191.4	302.4	388.5	499.3	724.7		
Cases	29	35	35	47	40		
Person-years	45,728	45,180	45,561	44,160	44,122		
Age-sex adjusted HR (95% CI)	1 (Ref.)	0.98 (0.60–1.61)	0.79 (0.48–1.30)	0.98 (0.61–1.57)	0.76 (0.47–1.21)	0.27	1.14 (0.76–1.72)
Multivariable adjusted model 1	1 (Ref.)	0.96 (0.54–1.71)	0.67 (0.38–1.22)	0.97 (0.55–1.69)	0.71 (0.4–1.24)	0.32	1.23 (0.76–1.98)
Restricted to >50 years old	1 (Ref.)	1.14 (0.62–2.11)	0.71 (0.37–1.34)	0.96 (0.52–1.78)	0.67 (0.37–1.2)	0.20	1.16 (0.69–1.95)
Repeated measurements model 1	1 (Ref.)	0.79 (0.45–1.40)	0.68 (0.38–1.18)	0.78 (0.45–1.37)	0.71 (0.41–1.26)	0.42	1.35 (0.85–2.14)
Restricted to >50 years old	1 (Ref.)	0.68 (0.38–1.22)	0.64 (0.35–1.15)	0.78 (0.43–1.39)	0.56 (0.31–1.00)	0.15	1.52 (0.95–2.44)
**Lignans**							
Median intake (mg/d)	1.2	1.7	2.1	2.5	3.4		
Cases	29	42	27	47	41		
Person-years	46,726	46,246	45,187	44,094	42,500		
Age-sex adjusted HR (95% CI)	1 (Ref.)	1.06 (0.66–1.70)	0.67 (0.40–1.14)	1.05 (0.65–1.68)	0.85 (0.53–1.36)	0.57	1.10 (0.73–1.65)
Multivariable adjusted model 1	1 (Ref.)	0.94 (0.55–1.61)	0.63 (0.35–1.146	0.92 (0.53–1.61)	0.78 (0.44–1.35)	0.47	1.20 (0.75–1.92)
Restricted to >50 years old	1 (Ref.)	0.57 (0.32–1.04)	0.65 (0.36–1.17)	0.77 (0.43–1.38)	0.67 (0.37–1.2)	0.50	1.53 (0.95–2.45)
Repeated measurements model 1	1 (Ref.)	0.90 (0.52–1.56)	0.68 (0.37–1.24)	0.95 (0.54–1.66)	0.81 (0.46–1.43)	0.67	1.19 (0.74–1.92)
Restricted to >50 years old	1 (Ref.)	0.51 (0.28–0.93)	0.64 (0.35–1.16)	0.98 (0.56–1.72)	0.59 (0.34–1.03)	0.53	1.52 (0.96–2.40)
**Phenolic acids intake**							
Median intake (mg/d)	147.7	221.9	285.7	352.5	482.4		
Cases	19	41	35	36	55		
Person-years	44,495	44,995	44,582	45,418	45,262		
Age-sex adjusted HR (95% CI)	1 (Ref.)	1.39 (0.80–2.40)	1.19 (0.68–2.09)	1.08 (0.61–1.88)	1.44 (0.82–2.53)	0.36	0.78 (0.48–1.26)
Multivariable adjusted model 1	1 (Ref.)	1.50 (0.79–2.83)	1.31 (0.69–2.49)	1.33 (0.7–2.51)	1.19 (0.63–2.25)	0.84	0.75 (0.43–1.32)
Restricted to >50 years old	1 (Ref.)	1.16 (0.64–2.12)	1.07 (0.57–1.99)	1.04 (0.56–1.94)	0.99 (0.52–1.84)	0.76	0.94 (0.56–1.55)
Repeated measurements model 1	1 (Ref.)	1.41 (0.75–2.68)	1.39 (0.74–2.64)	1.32 (0.70–2.51)	1.17 (0.62–2.22)	0.81	0.75 (0.43–1.33)
Restricted to >50 years old	1 (Ref.)	1.08 (0.60–1.96)	1.22 (0.67–2.20)	0.83 (0.44–1.55)	1.01 (0.54–1.90)	0.81	0.96 (0.60–1.57)
**Stilbenes intake**							
Median intake (mg/d)	−0.0	0.1	0.4	0.8	3.6		
N	3564	3564	3564	3564	3564		
Cases	29	26	31	40	60		
Person-years	45,603	44,073	44,221	44,409	46,446		
Age-sex adjusted HR (95% CI)	1 (Ref.)	1.02 (0.60–1.73)	0.84 (0.50–1.39)	0.83 (0.51–1.34)	0.75 (0.46–1.21)	0.20	1.20 (0.81–1.80)
Multivariable adjusted model 1	1 (Ref.)	0.91 (0.48–1.72)	0.58 (0.31–1.08)	0.84 (0.46–1.53)	0.67 (0.37–1.21)	0.31	1.38 (0.83–2.28)
Restricted to >50 years old	1 (Ref.)	0.58 (0.31–1.09)	0.75 (0.4–1.42)	0.64 (0.34–1.2)	0.51 (0.27–0.95) *	0.09	1.66 (0.99–2.79)
Repeated measurements model 1	1 (Ref.)	0.87 (0.47–1.64)	0.55 (0.29–1.02)	0.75 (0.42–1.37)	0.67 (0.39–1.17)	0.50	1.42 (0.86–2.32)
Restricted to >50 years old	1 (Ref.)	0.59 (0.32–1.07)	0.64 (0.34–1.21)	0.64 (0.35–1.18)	0.46 (0.25–0.85) *	0.08	1.75 (1.06–2.90) *
**Other phenolic compounds intake**							
Median intake (mg/d)	14.2	25.3	34.0	45.2	72.4		
Cases	34	33	29	40	50		
Person-years	46,187	45,737	45,103	43,789	43,936		
Age-sex adjusted HR (95% CI)	1 (Ref.)	0.88 (0.55–1.43)	0.80 (0.49–1.31)	1.07 (0.67–1.69)	1.11 (0.70–1.75)	0.33	1.03 (0.71- 1.5)
Multivariable adjusted model 1	1 (Ref.)	0.74 (0.42–1.29)	0.70 (0.39–1.25)	1.10 (0.64–1.89)	1.08 (0.63–1.87)	0.22	1.10 (0.7–1.72)
Restricted to >50 years old	1 (Ref.)	0.94 (0.49–1.82)	0.92 (0.47–1.84)	1.93 (1.04–3.59)	1.38 (0.74–2.58)	0.08	0.79 (0.46–1.33)
Repeated measurements model 1	1 (Ref.)	0.73 (0.42–1.28)	0.82 (0.46–1.45)	1.04 (0.61–1.80)	1.01 (0.59–1.74)	0.46	1.10 (0.71–1.72)
Restricted to >50 years old	1 (Ref.)	0.85 (0.45–1.62)	1.03 (0.55–1.96)	1.76 (0.96–3.23)	1.12 (0.61–2.06)	0.33	0.87 (0.52–1.45)

* *p* < 0.05. Abbreviations: Q: quintile; HR: hazard ratio; CI: confidence interval. All Cox regression models used were stratified by age (five-year periods), marital status, recruitment period and years of college education. Multivariable adjusted model 1: Additionally adjusted for total energy intake (kcal/day), BMI (kg/m^2^) and the quadratic term, smoking status (never smoker, current smoker or former smoker), lifetime tobacco exposure (packs-years), hypertension (yes/no), dyslipidemia(yes/no), family history of diabetes (yes/no), cardiovascular disease (yes/no), hormone replacement therapy (yes/no, only women) TV watching (hours/day), physical activity (metabolic equivalents-h/week, terciles), health consciousness (quintiles), energy-adjusted glycemic index (0–100) and energy-adjusted alcohol intake from sources other than wine. Repeated measurements model 1 were fitted for the same variables on model 1 with updated data at 10 years of follow-up (except TV watching, physical activity and health consciousness).

## Data Availability

The data that support the findings of this study are available from the SUN Project upon reasonable request at sun@unav.es.

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
