# Peer review of "Effect of Dietary Phenolic Compounds on Incidence of Type 2 Diabetes in the “Seguimiento Universidad de Navarra” (SUN) Cohort"

_antioxidants, 2023, doi:10.3390/antiox12020507_

Round 1

Reviewer 1 Report

Dear authors,

in my opinion the manuscript is suitable for the publication due its extensive analysis of dietary risk for T2D. Although the work appears complete, I think that is plausible insert in the study protocol an experimental approach in order to validate these data by biological manner (for example, measurements in biological fluids of antioxidant enzymes) for completion. 

It's clear that the work extensively studied many questionnaires attesting the use and intake of various dietary compound, but I'm wondering why the questionnaire did not reported the use of hormonal medication (such as the use of anticonceptionals, or other for menopause...) it could be very important for the correct identification of the risk.  Please, if you have these data, add in a table.

Reviewer 2 Report

The manuscript by Vázquez-Ruiz and collaborators aims to study the impact of dietary phenolic compounds on the risk of developing type 2 diabetes. This is a pertinent topic since the prevalence of type 2 diabetes continue to increase and it is important to find strategies to manage this disease. The beneficial effect of phenolic compounds on type 2 diabetes have already been described, even associated with Mediterranean diet. However, in the present manuscript the authors performed a follow-up of 10 years. The manuscript is well written and designed.  Comments are made to improve manuscript quality.

Comments:

-       I suggest distinguishing the present manuscript from the PREDIMED study. What is the novelty of the present manuscript?

-       In the present manuscript mean age of the participants are 37.4 years and by limiting the analysis for participants over 50 years old it was described significant results. Do you think that the consumption of food with higher phenolic compounds could be more beneficial for older people, that could have a higher predisposition to develop type 2 diabetes due to the age and other metabolic diseases that could develop, when compared with younger people? Since the participants are in the adult age this could explain the absence of significant results?
